# Acute Eosinophilic Pneumonia after Combined Use of Conventional and Heat-Not-Burn Cigarettes: A Case Report

**DOI:** 10.3390/medicina58111527

**Published:** 2022-10-26

**Authors:** Bo Hyoung Kang, Dong Hyun Lee, Mee Sook Roh, Soo-Jung Um, Insu Kim

**Affiliations:** 1Department of Internal Medicine, Dong-A University College of Medicine, Busan 49241, Korea; 2Department of Pathology, Dong-A University College of Medicine, Busan 49241, Korea

**Keywords:** bronchoalveolar lavage, cigarette smoking, electronic nicotine delivery system, eosinophilic pneumonia, steroid

## Abstract

*Background*: Acute eosinophilic pneumonia (AEP) is a rare acute respiratory disease accompanied by fever, shortness of breath, and cough. Although the pathogenesis of the disease is not yet established, the patient’s condition improves with a rapid therapeutic response to systemic corticosteroids. Conventional cigarettes or heat-not-burn cigarettes are the most common cause of AEP among young people. *Case Presentation*: A 22-year-old woman with dyspnea, cough, and fever did not improve after visiting the local medical center and was admitted to the emergency room. The patient denied having any recent travel history or insect bites. She was treated with appropriate antibiotics according to the community acquired pneumonia, but there was no improvement. Chest radiography showed bilateral patches of pulmonary infiltration, and chest computed tomography revealed bilateral multifocal patchy consolidations with multiple small nodular ground-glass opacities and interlobular septal thickening. The bronchoalveolar lavage result was dominantly eosinophilic. The patient’s condition improved rapidly after the use of intravenous methylprednisolone and then a change to oral methylprednisolone. Finally, the patient was hospitalized for 9 days, and the duration of use of methylprednisolone including outpatient visits was 14 days. *Results*: The early treatment of AEP yields a good prognosis, but since the symptoms of AEP are similar to those of infectious diseases such as community-acquired pneumonia, physicians should be meticulous in differentiating AEP from other diseases. *Conclusions*: Since AEP shows a good response to steroids, early detection using an appropriate diagnostic method is recommended. In addition, there should be strong education against smoking in any form.

## 1. Introduction

Acute eosinophilic pneumonia (AEP) refers to pulmonary eosinophilia that occurs within a few days to several weeks and is accompanied by nonspecific symptoms such as cough, dyspnea, and occasionally malaise and myalgia [1]. Since AEP is a rare disease, the exact incidence and cause are not clear, but it is reported that the incidence is higher in young people or those who first start smoking [2]. AEP has non-specific symptoms such as cough, shortness of breath, and fever, so it is difficult to differentiate it from other infectious diseases [1]. It is helpful for diagnosis to perform bronchoalveolar lavage (BAL) after excluding other diseases using a well-ordered questionnaire that includes a travel history and medication history [1]. AEP improves when steroids are used, and the prognosis is good, so it is important to make an appropriate diagnosis early [3].

Heat-Not-Burn cigarettes (HNBCs) utilize a method of inhaling vaporized aerosol by heating the cigarette using an electric device. And HNBCs is often used by first-time smokers or those who are trying to quit smoking [4]. The Republic of Korea’s government also accepted and signed the World Health Organization framework convention on tobacco control in 2003 and strengthened anti-smoking activities in the public and private sectors, so the overall smoking rate decreased [5]. However, since HNBCs were introduced in Korea in June 2017, sales have gradually increased, accounting for about 10.5% of the total cigarette sales in 2019 [5]. Manufacturers have been promoting HNBCs as a less harmful alternative to conventional cigarettes (CCs) for humans because they contain fewer toxic substances [6]. However, a recent study found that the nicotine content was similar to that of CCs, and other harmful ingredients, such as formaldehyde, were also detected; thus, HNBCs is not less harmful than CCs [6]. In addition, the detrimental effects of HNBCs on the human body have not yet been clearly identified [7].

## 2. Case Presentation

A previously healthy 22-year-old female presented with dyspnea, cough, and fever in a local hospital 1 day prior to visiting. On chest radiography, diffuse infiltration was observed in both lung fields, and broad-spectrum antibiotics were administered under the suspicion of community-acquired pneumonia. However, the symptoms did not improve, hence her referral to our hospital for further evaluation and treatment.

The patient denied having any recent travel history or insect bites. She has no allergic disease such as asthma or atopy and has never used illegal drugs. She had never smoked before. However, 2 weeks before the onset of symptoms, she learned to smoke from her boyfriend due to the stress of her studies, and since she was reluctant to smoke CCs, she started smoking HNBCs. She initially started with 6 cigarettes per day but increased to 15 cigarettes per day just before the onset of symptoms. On admission, the patient’s vital signs were as follows: body temperature, 38.2 °C; respiratory rate, 30 breaths/min; heart rate, 102 beats/min; and oxygen saturation, 93% on 3-L oxygen via nasal cannula. An initial complete blood cell count showed an elevated white blood cell count of 24.4 × 103/μL with 96.7% neutrophils, 0.9% eosinophils, and 1.2% lymphocytes. Crackles were detected in the bilateral lung fields on auscultation. Chest radiography showed bilateral patches of pulmonary infiltration (Figure 1A), and chest computed tomography revealed bilateral multifocal patchy consolidations with multiple small nodular ground-glass opacities and interlobular septal thickening (Figure 1B).

BAL was performed for an accurate diagnosis. In total, 2.7 × 103/mm^3^ cells comprising 62% eosinophils, 15% lymphocytes, 14% macrophages, and 4% neutrophils were withdrawn from the BAL fluid. The patient’s condition improved rapidly after the use of intravenous methylprednisolone. The patient was hospitalized for a total of 9 days, and on the fifth day of admission, the symptoms improved and the patient was changed to oral methylprednisolone (62.5 mg to 40 mg). After that, her symptoms continued to improve, and the dose was reduced from 40 mg to 20 mg on the 10th day of steroid administration.

After discharge, on the outpatient visit, there were no symptoms and the imaging test results were normal (Figure 2A,B). The methylprednisolone was stopped on the 14th day after starting the administration. On follow-up 6 months after she was discharged, she was no longer smoking, and there was no recurrence of AEP.

## 3. Discussion

AEP is a rare disease known to have various causes, though the most prevalent one is smoking [2]. AEP develops more frequently in populations of heavy smokers (e.g., military personnel) than in the general population [8], and it is found more often in young people who start smoking for the first time [2]. With the development of electronic technology, new cigarettes (e.g., E-cigarettes, HNBCs), aside from the CCs, have been made more accessible to first-time smokers or women; thus, unlike the general smoking population, the smoking rate of these subpopulations has been increasing [4,5,7], and side effects such as AEP are often reported [9,10]. Smoking is the most common cause of AEP, but illegal drugs, medications, and the occupational environment are also often a cause of AEP [11,12,13], so it is necessary to differentiate it before the diagnostic test through a well-organized questionnaire [1].

Clinical features of AEP include rapidly progressive dyspnea, cough, and chest discomfort. Since AEP is difficult to differentiate from community-acquired pneumonia and other inflammatory diseases, in the diagnosis of AEP, it is vital to consider the patient’s symptoms and history including the use of illegal drugs and medications [1]. Recent bilateral lung infiltration found on chest radiography also aids in the diagnosis [14]. In addition, parenchymal infiltration should be present in the chest image, but the most important thing for diagnosis is to find eosinophils of 25% or more in BAL specimen [1,15].

In this case, our patient fulfilled all of the criteria. Although eosinophilia in peripheral blood can be helpful in determining the diagnosis, it is essential to differentiate it from parasitic infections or autoimmune diseases [1]. According to a study by Giacomi et al., smoking-related AEP is less associated with peripheral eosinophilia than medication-related AEP [2]. In the study of Suzuki et al., only 30% of AEP patients showed peripheral eosinophilia, and peripheral eosinophilia is not an essential indicator for the diagnosis of AEP [15]. Therefore, eosinophilia observed in BAL fluid is more definitive; thus, it is recommended that patients with suspected AEP undergo BAL promptly [1,15]. Glucocorticoid therapy is the mainstay treatment for AEP, but to date, there is no clear consensus on the appropriate duration and dose of administration [1,3].

In general, after the diagnosis of AEP is made and glucocorticoid treatment is initiated, patients’ symptoms rapidly improve, and the prognosis of AEP is excellent [1]. Therefore, proper diagnosis and administration of glucocorticoids are crucial [3].

Several studies found that e-cigarettes and HNBCs are as harmful as CCs [6,7], and the simultaneous use of conventional and e-cigarettes or HNBCs is more dangerous than using any of the products alone [16]. In this case, it is unclear whether CCs or HNBCs influenced the onset of AEP, but it is believed that a synergistic effect of the two kinds was generated.

## 4. Conclusions

Although this patient exhibited good outcomes due to proper diagnosis and prompt treatment, AEP is often mistaken for community-acquired pneumonia. Therefore, when diagnosing a patient with symptoms, such as dyspnea, dry cough, and diffuse pulmonary infiltration on radiographs, physicians should carefully distinguish AEP from other diseases by examining the history of smoking or drug use. When differential diagnosis is not possible with a medical examination by interview alone, it is necessary to utilize interventions such as BAL for an early diagnosis.

## Figures and Tables

**Figure 1 medicina-58-01527-f001:**
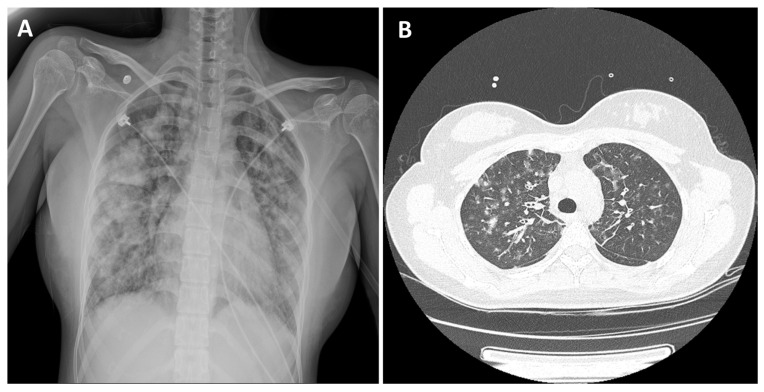
Initial imaging findings of a patient with acute eosinophilic pneumonia. (**A**) Chest radiograph revealing a bilateral patch of pulmonary infiltration. (**B**) Chest computed tomography revealing bilateral multifocal patchy consolidations with multiple small nodular ground-glass opacities and smooth interlobular septal thickening.

**Figure 2 medicina-58-01527-f002:**
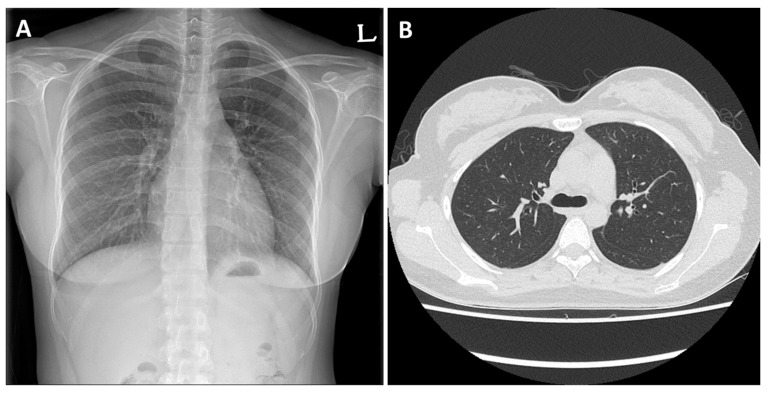
(**A**) Chest radiograph and (**B**) chest computed tomography showing improvement of bilateral infiltration and multiple small nodular ground-glass opacities after treatment.

## Data Availability

Not applicable.

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
