# Peer review of "Acute Eosinophilic Pneumonia after Combined Use of Conventional and Heat-Not-Burn Cigarettes: A Case Report"

_medicina, 2022, doi:10.3390/medicina58111527_

Round 1

Reviewer 1 Report

  Acute eosinophilic pneumonia that was presented as a case report in this manuscript presents and, in my opinion, will provide a valuable source document for anyone requiring a primer to know and understand this issue. Some changes are needed:    

  • Lines 14-16: Consider avoiding writing in the first person way.  
  • Lines 34-42: The first paragraph in the section Introduction must be reconstructed with the goal of a clearer presentation of basic information regarding acute eosinophilic pneumonia: definition, epidemiology, etiology (`cause`, ie risk factors), clinical features, diagnosis, treatment, prognosis.  
  • Throughout the entire manuscript: For every sentence that is a claim/information from the literature it is necessary to cite the appropriate reference. It is important to cite the reference and the end of the first sentence where you for the first time provide information from the literature. 
  • Line 47: State whether and when the Republic of Korea accepted and signed `The WHO Framework Convention on Tobacco Control`.  
  • Lines 53-55: Precisely state the location (country, region, world) to which the following claim refers: `to date, there is no report on AEP that developed after using CCs and HNBC simultaneously.`.  
  • Lines 55-57: Consider avoiding writing in the first person way.  
  • Line 64: In the paper, describe personal medical history that is potentially important for the occurrence of the described disease (e.g. atopy or asthma, etc) at any point in time. In any case, information about this must be stated, even if the person was completely healthy, without any diseases, conditions or injuries in their personal medical history. 
  • Lines 64-67: In line with the goal of this paper, the description of the smoking status must be explained in a more detailed and clear way: when did the person start smoking cigarettes for the first time in their life, how many cigarettes did they smoke on average per day, what is the length of their smoking habit - how long have they been smoking, have they stopped smoking, have they inhaled the smoke. Also, if the person has never before smoked any cigarettes, clearly state that the person has for the first time started smoking and immediately started simultaneously smoking two types of cigarettes. It is important to provide information whether there have been any smokers in the patient's surrounding, both for conventional and heat-not-burn cigarettes, whether it is someone close, household or work contact. 
  • Line 87: Add a new paragraph that would state the current diagnostic criteria for acute eosinophilic pneumonia, with citation of the appropriate reference. 
  • Lines 107-109: Reference No 11 is from 2005, have the diagnostic criteria for acute eosinophilic pneumonia remained the same or have some other authors updated and published it. Cite the reference.   
  • Lines 111-113: It is not the best practice to cite a case report (ref No 3) for the claim `thus, it is recommended that patients with suspected AEP undergo BAL promptly [3].`. For such a claim, an appropriate reference should be cited. 
  • Line 123: Conclusions?   

Author Response

Thank you for inviting us to revise and re-submit our manuscript for publication in Medicina. We have addressed the concerns raised by the editor in a point-by-point format. The English has been professionally checked again before resubmitting this article. All changes have been marked in red font in the revised manuscript. We hope that the changes meet with your approval.

# Response to the comments of Reviewer 1

▪ Comment 1

 Lines 14-16: Consider avoiding writing in the first person way. 

▪ Response

We thank the reviewer for bringing this point to our attention. As you pointed out, the sentence was deleted as it was deemed unnecessary.

▪ Comment 2

Lines 34-42: The first paragraph in the section Introduction must be reconstructed with the goal of a clearer presentation of basic information regarding acute eosinophilic pneumonia: definition, epidemiology, etiology (`cause`, ie risk factors), clinical features, diagnosis, treatment, prognosis. 

Throughout the entire manuscript: For every sentence that is a claim/information from the literature it is necessary to cite the appropriate reference. It is important to cite the reference and the end of the first sentence where you for the first time provide information from the literature.

▪ Response

We thank the reviewer for bringing this point to our attention. As you said, the first paragraph of the introduction was restructured, and the contents were added while deleting inappropriate contents. In addition, citations were rechecked in all manuscripts, and missing citations were added. We have changed paragraph in the introduction section as follows (see page 1, line 34-43): “AEP refers to pulmonary eosinophilia that occurs within a few days to several weeks and is accompanied by non-specific symptoms such as cough, dyspnea, and rarely malaise and myalgia [1]. Since AEP is a rare disease, the exact incidence and cause are not clear, but it is reported that the incidence is higher in young people or those who first start smoking [2]. AEP has non-specific symptoms such as cough, shortness of breath, and fever, so it is difficult to differenti-ate it from other infectious diseases [1]. It is helpful for diagnosis to perform BAL after excluding other diseases using a well-ordered questionnaire that includes travel history and medication history [1]. AEP improves when steroids are used, and the prognosis is good, so it is important to make an appropriate diagnosis early [3].”

▪ Comment 3

Line 47: State whether and when the Republic of Korea accepted and signed `The WHO Framework Convention on Tobacco Control`.

▪ Response

We thank the reviewer for bringing this point to our attention. The Republic of Korea signed the WHO framework convention on tobacco control in 2003, and is strengthening anti-smoking campaigns in the private and public sectors. We have added information about Korea's WHO smoking cessation agreement and changed paragraph in the introduction section as follows (see page 2, line 46-50): “The Republic of Korean government also accepted and signed the WHO framework convention on tobacco control in 2003, and strengthened anti-smoking activities in the public and private sectors, so the overall smoking rate decreased [5]. However, since HNBC was introduced in Korea in June 2017, sales have gradually increased, accounting for about 10.5% of total cigarette sales in 2019 [5].”

▪ Comment 4

Lines 53-55: Precisely state the location (country, region, world) to which the following claim refers: `to date, there is no report on AEP that developed after using CCs and HNBC simultaneously.`

▪ Response

We thank the reviewer for bringing this point to our attention. As you pointed out, unnecessary sentences in the entire context were deleted.

▪ Comment 5

Lines 55-57: Consider avoiding writing in the first person way. 

▪ Response

We thank the reviewer for bringing this point to our attention. As you pointed out, inappropriately expressed sentences have been deleted.

▪ Comment 6

Line 64: In the paper, describe personal medical history that is potentially important for the occurrence of the described disease (e.g. atopy or asthma, etc) at any point in time. In any case, information about this must be stated, even if the person was completely healthy, without any diseases, conditions or injuries in their personal medical history.

▪ Response

We thank the reviewer for bringing this point to our attention. Past history closely related to AEP could not be described in detail. Medical records were reviewed and information on past history was added in the case presentation section as follows (see page 2, line 57):“A previously healthy.”

(see page 2, line 62-63): “She has no allergic disease such as asthma or atopy, and has never used illegal drugs.”

▪ Comment 7

Lines 64-67: In line with the goal of this paper, the description of the smoking status must be explained in a more detailed and clear way: when did the person start smoking cigarettes for the first time in their life, how many cigarettes did they smoke on average per day, what is the length of their smoking habit - how long have they been smoking, have they stopped smoking, have they inhaled the smoke. Also, if the person has never before smo ked any cigarettes, clearly state that the person has for the first time started smoking and immediately started simultaneously smoking two types of cigarettes. It is important to provide information whether there have been any smokers in the patient's surrounding, both for conventional and heat-not-burn cigarettes, whether it is someone close, household or work contact.

▪ Response

We thank the reviewer for bringing this point to our attention. In the initial menuscript, the patient's occupation and department, and what he learned to smoke from his boyfriend, was deleted after being advised to delete it due to excessive exposure of personal information during the English proofreading process. She was a college student in her 3rd year of architecture. She started learning tobacco from her boyfriend due to the stress of producing her graduation work, and said that she had a reluctance to smoke from the beginning, so she used HNBC together. Based on the patient's statements, information on smoking patterns was added in the case presentation section as follows (see page 2, line 63-67): “She had never smoked before. However, 2 weeks before the onset of symptoms, she learned to smoke from her boyfriend due to the stress of her studies, and she was reluctant to do CC, so she started HNBC together. She initially started with 6 cigarettes per day, but increased to 15 cigarettes per day just before the onset of symptoms.”

▪ Comment 8

Line 87: Add a new paragraph that would state the current diagnostic criteria for acute eosinophilic pneumonia, with citation of the appropriate reference.

▪ Response

We thank the reviewer for bringing this point to our attention. A number of diagnostic criteria have been proposed since allen et al. in 1989 made a distinct classification for AEP. According to a study by Giacomi et al. in 2018, respiratory symptoms that worsen in a short period of time and parenchymal infiltraion on imaging should be present, and infectious diseases such as CAP should be excluded. The most important thing in the diagnosis of AEP is that the cause should be identified through a systematic questionnaire, and more than 25% of eosinophils in BAL should be found, and appropriate clinical and imaging characteristics should be accompanied. Therefore, a paragraph has been added in discussion to reflect the recently presented diagnostic criteria of AEP (see page 3, line 109-115): “Clinical features of AEP include rapidly progressive dyspnea, cough, and chest dis-comfort. Since AEP is difficult to differentiate from CAP and other inflammatory diseases, in the diagnosis of AEP, it is vital to consider the patient's symptoms and history including illegal drug and medications [1]. Recent bilateral lung infiltration found on chest radiography also aids in the diagnosis [14]. In addition, parenchymal infiltration should be present in the chest image, but the most important thing for diagnosis is to find eosinophils of 25% or more in BAL specimen [1].”

▪ Comment 9

Lines 107-109: Reference No 11 is from 2005, have the diagnostic criteria for acute eosinophilic pneumonia remained the same or have some other authors updated and published it. Cite the reference.

▪ Response

We thank the reviewer for bringing this point to our attention. We added recent findings on the diagnosis of acute eosinopilic pneumonia, and revised the reference in the discussion section as follows (see page 3, line 109-115): “Clinical features of AEP include rapidly progressive dyspnea, cough, and chest dis-comfort. Since AEP is difficult to differentiate from CAP and other inflammatory diseases, in the diagnosis of AEP, it is vital to consider the patient's symptoms and history including illegal drug and medications [1]. Recent bilateral lung infiltration found on chest radiography also aids in the diagnosis [14]. In addition, parenchymal infiltration should be present in the chest image, but the most important thing for diagnosis is to find eosinophils of 25% or more in BAL specimen [1].”

▪ Comment 10

Lines 111-113: It is not the best practice to cite a case report (ref No 3) for the claim `thus, it is recommended that patients with suspected AEP undergo BAL promptly [3].`. For such a claim, an appropriate reference should be cited.

▪ Response

We thank the reviewer for bringing this point to our attention. The reference presented in the case report was inappropriate. Therefore, the reference has been changed to a review article. as follows (see page 4, line 122-123): “Therefore, eosinophilia observed in BAL fluid is more definitive; thus, it is recommended that patients with suspected AEP undergo BAL promptly [1,15].”

▪ Comment 11

Line 123: Conclusions?  

▪ Response

We thank the reviewers for pointing this out and correcting it. There was a mistake in writing the document, so it was not properly labeled. I have corrected it as you pointed out.

Reviewer 2 Report

The work presented to me for the review has a significant practical importance. The interesting case of AEP resulting in the following use of electronic cigarettes. - emphasizes the value of a well -collected interview in diagnostics and also the value of performing a  BAL in such cases.

Author Response

Thank you for inviting us to revise and re-submit our manuscript for publication in Medicina. We have addressed the concerns raised by the editor in a point-by-point format. The English has been professionally checked again before resubmitting this article. All changes have been marked in red font in the revised manuscript. We hope that the changes meet with your approval.

# Response to the comments of Reviewer 2

▪ Comment

 The work presented to me for the review has a significant practical importance. The interesting case of AEP resulting in the following use of electronic cigarettes. - emphasizes the value of a well -collected interview in diagnostics and also the value of performing a BAL in such cases.

▪ Response

We thank the reviewer for bringing this point to our attention. As you said, several literatures also emphasize the importance of patient history. The patient's past history includes not only medication, cigarette smoking, occupation, travel, illegal drug, but also experiences of special disasters such as the collapse of the World Trade Center due to the 9/11 terrorist attacks. However, not all patients experience this and there is a time limit for emergency room visits, so it is necessary to focus on the most commonly suspected causes by region and age. Therefore, basic factors such as smoking history and drug taking history should be asked from all patients, and related questions should be asked in areas where access to illegal drugs is easy. In addition, in special situations such as overseas dispatch, you should ask questions about factors such as parasites or virus infection as well as the above items.

Therefore, we added the content about the importance of questionnaires to the Discussion part as follows. (see page 3, line 105-108): “Smoking is the most common cause of AEP, but illegal drug, medications, and occupa-tional environment are also often a cause of AEP [11-13], so it is necessary to differentiate it before the diagnostic test through a well-organized questionnaire [1].”

Reviewer 3 Report

In the Abstract it is written that the methylprednisolone was administered 14 day after hospitalization, but is not written how many day the patient stayed in hospital. I suggest to input the number of days of the whole steroid therapy.

 In the text about Case report it is written that initial results showed that the patient has 1.2% lymphocytes – is it a mistake? This is not an expected value for  lymphocytes. 

Also there is no information about the dosage of methylprednisolone.

 Why there are two Sections with the same title: 3. Discussion and 4. Discussion? Probably the title for the number 4. should be Conclusion.

 I would suggest more comments on lack of peripheral eosinophilia in this Case, in the section Discussion.

Author Response

Thank you for inviting us to revise and re-submit our manuscript for publication in Medicina. We have addressed the concerns raised by the editor in a point-by-point format. The English has been professionally checked again before resubmitting this article. All changes have been marked in red font in the revised manuscript. We hope that the changes meet with your approval.

# Response to the comments of Reviewer 3

▪ Comment 1

 In the Abstract it is written that the methylprednisolone was administered 14 day after hospitalization, but is not written how many day the patient stayed in hospital. I suggest to input the number of days of the whole steroid therapy.

▪ Response

We thank the reviewer for bringing this point to our attention. It seems that the patient's medication history was written in a confusing way. The patient's steroid prescription history and hospitalization history will be clearly described. Therefore, we changed the paragraph about the administration history of medications to the abstract and case presentation part as follows (see page 1, line 22-24): “The patient was hospitalized for 9 days, and the duration of use of methylprednisolone including outpatient visits was 14 days.”

(see page 2, line 85-96): “The patient was hospitalized for a total of 9 days, and on the 5th day of admission, the symptoms improved and the patient was changed to oral methylprednisolone (62.5mg to 40mg). After that, symptoms continued to improve, and the dose was reduced from 40 mg to 20 mg on the 10th day of steroid administration. After discharge, on the outpatient visit, no symptoms and the imaging test results were normal (Fig. 2-A,B)., and the methylprednisolone was stopped on the 14th day after starting the administration. On follow-up 6 months after she was discharged, she was no longer smoking, and there was no recurrence of AEP.”

▪ Comment 2

 In the text about Case report it is written that initial results showed that the patient has 1.2% lymphocytes – is it a mistake? This is not an expected value for lymphocytes.

▪ Response

We thank the reviewer for bringing this point to our attention. In medical record, this patient had severe neutrophilia at the initial diagnosis of acute eosinophilic pneumonia, and as mentioned, the fraction of lymphocytes was very low. The neutrophil and lymphocyte fractions on the CBC of this patient were summarized and graphed. In this patient, the fraction of lymphocytes was normalized on the 7th day of steroid administration.

However, reviewing several literatures, an increase in the fraction of lymphocytes on CBC is not necessarily associated with acute eosinophilic pneumonia patients. In the diagnosis of acute eosinophilic pneumonia, an increase in ESR or IgE levels may be helpful in the diagnosis, but this patient did not perform it. If a test was performed before steroid administration, it is thought that the diagnosis was helpful.

References

  1. Tuberc Respir Dis (Seoul). 2013 Feb; 74(2): 51–55.
  2. Am J Respir Crit Care Med. 2018 Mar 15;197(6):728-736.

▪ Comment 3

Also there is no information about the dosage of methylprednisolone.

▪ Response

We thank the reviewer for bringing this point to our attention. we added the content about the dosage of steroid to the case presentation part as follows (see page 2, line 85-89): “The patient was hospitalized for a total of 9 days, and on the 5th day of admission, the symptoms improved and the patient was changed to oral prednisolone (62.5mg to 40mg). After that, symptoms continued to improve, and the dose was reduced from 40 mg to 20 mg on the 10th day of steroid administration.”

▪ Comment 4

 Why there are two Sections with the same title: 3. Discussion and 4. Discussion? Probably the title for the number 4. should be Conclusion.

▪ Response

We thank the reviewers for pointing this out and correcting it. There was a mistake in writing the document, so it was not properly labeled. I will correct it as you pointed out.

▪ Comment 5

 I would suggest more comments on lack of peripheral eosinophilia in this Case, in the section Discussion.

▪ Response

We thank the reviewer for bringing this point to our attention. Peripheral eosinophilia is more common in drug-induced pneumonia, and may or may not occur in smoking-induced acute eosinophilic pneumonia. Therefore, acute eosinophilic pneumonia caused by smoking is not necessarily accompanied by peripheral eosinophilia, and the presence of peripheral eosinophilia should not be used to determine whether acute eosinophilic pneumonia is present. We have added to the discussion part about acute eosinophilic pneumonia diagnosis and peripheral eosinophilia as follows (see page 3, line 118- page 4, line 122): “According to a study by Giacomi et al., smoking-related AEP is less associated with pe-ripheral eosinophilia than medication-related AEP [2]. In the study of Suzuki et al., only 30% of AEP patients showed peripheral eosinophilia, and peripheral eosinophilia is not an essential indicator for the diagnosis of AEP [15].”

Round 2

Reviewer 1 Report

Thank you for the opportunity to re-review the manuscript ID-healthcare-1954497. The authors have addressed all of the issues highlighted in my review. The revised manuscript is clear, readable and informative and will provide valuable findings for this issue. Thank you to the authors for their responses to my comments. Short note/suggestion for authors: It would be good to define each abbreviation (AEP, BAL, WHO, ...) in the text of the paper (from Introduction throughout), or to do it when editing the text.